# Tenrec: A Large-scale Multipurpose Benchmark Dataset for Recommender Systems

**Guanghu Yuan**[1,3,4*], **Fajie Yuan**[2*], **Yudong Li**[1], **Beibei Kong**[1], **Shujie Li**[4], **Lei Chen**[3],
**Min Yang**[3], **Chenyun Yu**[5], **Bo Hu**[1], **Zang Li**[1], **Yu Xu**[1], **Xiaohu Qie**[1]

[1]Tencent    [2]Westlake University
[3]Shenzhen Institute of Advanced Technology, Chinese Academy of Sciences
[4]University of Science and Technology of China    [5]Sun Yat-sen University
gh.yuan0@gmail.com, yuanfajie@westlake.edu.cn, ustclsj@mail.ustc.edu.cn
{lei.chen,min.yang}@siat.ac.cn, yuchy35@mail.sysu.edu.cn
{elsonli,echokong,harryyfhu,gavinzli, henrysxu,tigerqie}@tencent.com

## Abstract

Existing benchmark datasets for recommender systems (RS) either are created at a small scale or involve very limited forms of user feedback. RS models evaluated on such datasets often lack practical values for large-scale real-world applications. In this paper, we describe Tenrec[*], a novel and publicly available data collection for RS that records various user feedback from four different recommendation scenarios. To be specific, Tenrec has the following five characteristics: (1) it is large-scale, containing around 5 million users and 140 million interactions; (2) it has not only positive user feedback, but also true negative feedback (vs. one-class recommendation); (3) it contains overlapped users and items across four different scenarios; (4) it contains various types of user positive feedback, in forms of clicks, likes, shares, and follows, etc; (5) it contains additional features beyond the user IDs and item IDs. We verify Tenrec on ten diverse recommendation tasks by running several classical baseline models per task. Tenrec has the potential to become a useful benchmark dataset for a majority of popular recommendation tasks. Our source codes, datasets and leaderboards are available at https://github.com/yuangh-x/2022-NIPS-Tenrec[*].

## 1 Introduction

Recommender systems (RS) aim to estimate user preferences on items that users have not yet seen. Progress in deep learning (DL) has spawned a wide range of novel and complex neural recommendation models. Many improvements have been achieved in prior literature, however, much of them performs evaluation on non-benchmark datasets or on datasets at a small scale by modern standard. This has led to severe reproducibility and credibility problems in the RS community. For example, [34] showed that many 'advanced' baselines reported in previous papers are largely suboptimal, even underperforms the vanilla matrix factorization (MF) [21], an old baseline proposed over a decade ago. [33] further demonstrated that with a careful setup dot product is superior to the

---

[*]Equal contribution. Fajie designed the research, Guanghu performed the research; Fajie, Guanghu, Lei and Min wrote the paper; Fajie, Min, Yudong and Beibei launched the research project; Guanghu, Beibei and Yudong collected the data; Shujie assisted in performing partial experiments. Experiments of this work were mainly performed when Guanghu interned at Tencent.

[*]Tenrec (a hedgehog-like mammal) here means that the dataset is collected from the recommendation platforms of Tencent, and that it can be used to benchmark ten diversified recommendation tasks.

[*]Email Fajie & Guanghu if you want to launch a new leaderboard for an important RS task using Tenrec.

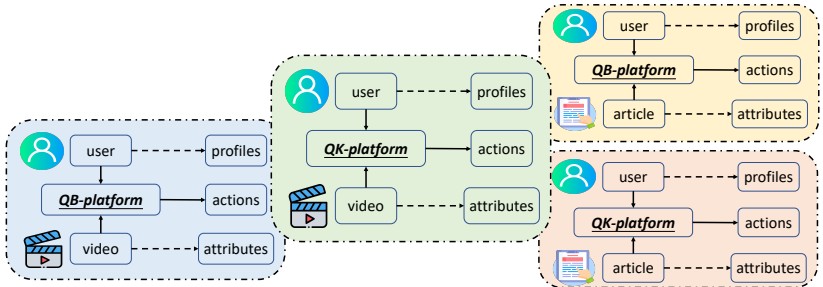

Figure 1: Data sources of Tenrec. Tenrec contains four recommendation scenarios: QK-video, QK-article, QB-video, and QB-article, where there are some percentage of overlapped users and items between every two or three scenarios.

learned similarity, e.g. using a multilayer perceptron (MLP) [15]. Recently, [9, 22, 41, 18, 8, 7, 1] also questioned some recognized progress in RS from the dataset and experimental setup perspectives.

A large-scale and high-quality datasets have a significant impact on accelerating research in an area, such as ImageNet [11] for computer vision (CV) and GLUE [44] for natural language processing (NLP). However, it is often difficult for researchers to access large-scale real-world datasets for studying recommendation problems due to security or privacy issues. Despite that, there still exist several popular datasets for regular recommendation tasks. For example, movieLens* (ML) datasets, including ML-100K, ML-1M and ML-10M, etc have become the stable benchmark datasets for the rating prediction [21] task. Other popular datasets, including Netflix [30] for movie recommendation, Yelp [6] for location recommendation, Amazon [14] for product recommendation, Mind [49] for news recommendation, Last.fm [56] and Yahoo! Music [13] for song recommendation, also appear frequently in literature. One major drawback of these datasets is that user feedback data has very limited forms, for example, most of them include only one type of user feedback (either rating, or clicking, or watching), or are collected from only one recommendation scenario. This severely limits the research scope of real-world recommender systems.

To foster diverse recommendation research, we propose Tenrec, a large-scale and multipurpose real-world dataset. Compared with existing public datasets, Tenrec has several merits: (1) it consists of overlapped users/items from four different real-world recommendation scenarios, which can be used to study the cross-domain recommendation (CDR) and transfer learning (TF) methods; (2) it contains multiple types of positive user feedback (e.g. clicks, likes, shares, follows, reads and favorites), which can be leveraged to study the multi-task learning (MTL) problem; (3) it has both positive user feedback and true negative feedback, which can be used to study more practical click-through rate (CTR) prediction scenario; (4) it has additional user and item features beyond the identity information (i.e. user IDs and item IDs), which can be used for context/content-based recommendations.

Owing to these advantages, Tenrec can be employed to evaluate a wide range of recommendation tasks. In this paper, we examine its properties by ten recommendation tasks, including (1) CTR prediction [62, 63], (2) session-based recommendation [16], (3) MTL recommendation [28], (4) CDR recommendation [61], (5) user profile prediction [55], (6) cold-start recommendation [43], (7) lifelong user representation learning [57], (8) model compression [40], (9) model training speedup [45], and (10) model inference speedup [2]. Beyond these tasks, we can easily integrate some of the above characteristics to propose additional or new tasks. To the best of our knowledge, Tenrec is so far one of the largest datasets for RS, covering a majority of recommendation scenarios and tasks. We release all datasets and codes to promote reproducibility and advance new recommendation research.

## 2 Dataset Description

Tenrec is a dataset suite developed for multiple recommendation tasks, collected from two different feeds recommendation platforms of Tencent*, namely, QQ BOW (QB) and QQ KAN (QK).* An item

---

*https://grouplens.org/datasets/movielens/

*https://www.tencent.com/en-us/

*The names of the two production systems have been anonymized (email us if you want to know their names).

Table 1: Data Statistics. avg #clicks denotes the average number of clicks per user; #exposure denotes the number of exposures to users (a.k.a. impressions), including both positive and negative feedback.

| Name | QK-video | QK-article | QB-video | QB-article |
|---|---|---|---|---|
| #users | 5,022,750 | 1,325,838 | 34,240 | 24,516 |
| #items | 3,753,436 | 220,122 | 130,637 | 7,355 |
| #click | 142,321,193 | 46,111,728 | 1,701,171 | 348,736 |
| #like | 10,141,195 | 821,888 | 20,687 | / |
| #share | 1,128,312 | 591,834 | 2,541 | / |
| #follow | 857,678 | 62,239 | 2,487 | / |
| #read | / | 44,228,593 | / | / |
| #favorite | / | 316,627 | / | / |
| #exposure | 493,458,970 | / | 2,442,299 | / |
| avg #clicks | 28.34 | 34.78 | 49.69 | 14.22 |

in QK/QB can either be a news article or a video. Note that the article and video recommendation models are trained separately with different neural networks and features. Thus, we can think that Tenrec is composed of user feedback from four scenarios in total, namely, QK-video, QK-article, QB-video, and QB-article (see Figure 1). We collect user behavior logs from QK/QB from September 17 to December 07, 2021. The procedure is as follows: we first randomly draw around 5.02 million users from the QK-video database, with the requirement that each user had at least 5 video clicking behaviors; then, we extract their feedback (around 493 million), including both positive feedback (i.e. video click, share, like and follow) and negative feedback (with exposure but no user action); finally, we obtain around 142 million clicks, 10 million likes, 1 million shares and 0.86 million follows, alongside 3.75 million videos. Besides, there are age and gender features for users, and the video type feature for items. We perform similar data extraction strategy for QK-article, QB-video, and QB-article. In this paper, we regard QK-video as the main scene, and other three as the secondary scenes, used for various CDR or TF tasks. The dataset statistics are shown in Table 1.

**Data Distribution.** Figure2 (a) and (b) show the item popularity of QK-video in terms of the clicking behaviors. Clearly, the item popularity follows a typical long-tail distribution, which has been widely reported in previous recommendation literature [31]. (c) shows the session length distribution, where the number of sessions with length in $[0 - 20]$ accounts for 53% of all sessions. Similar distributions can be observed on the other three datasets, which are thus simply omitted.

**Data Overlapping.** Tenrec contains a portion of overlapped users and items across the four scenarios. Regarding overlapped users, we calculate them between QK-video and QK-article, QB-video, QB-article given that QK-video covers the largest number of users, items and interactions. Specifically, the number of overlapped users is 268,207 between QK-video and QK-article, 3,261 between QK-video and QB-video, and 58 between QK-video and QB-article. Regarding overlapped items, 78,482 videos are overlapped between QK-video and QB-video. Overlapped users and items can be associated by their unique IDs. This property makes Tenrec well-suited for studying the TF and CDR tasks.

**User Feedback.** Tenrec is different from existing recommendation datasets, containing only one type of user feedback, either implicit feedback or explicit ratings. As a result, the degree of user preference in these datasets cannot be well reflected. As shown in Table 1, QK-video and QB-video include four types of positive feedback, where clicking behaviors account for the largest number, followed by likes, shares and follows. This finding is intuitive since likes, shares and follows often denote much higher preference than clicks. Likewise, QK-article contains two additional types of preference feedback, i.e. the article reading and favorite behaviors. Beyond various positive feedback, Tenrec includes true negative feedback — i.e. an item is present to a user, but s/he has not clicked it. Such negative feedback enables Tenrec to be more suited to CTR prediction, for which most existing datasets involve only positive feedback.

**Features.** The format of each instance in QK/QB-video is {*user ID, item ID, click, like, share, follow, video category, watching times, user gender, user age*}. Note that the timestamp information has been removed required by Tencent, but we present all interaction behaviors according to the time order. *click, like, share, follow* are binary values denoting whether the user has such an action. *watching times* is the number of watching behaviors on the video. *user ID, item ID, user gender, user age* have

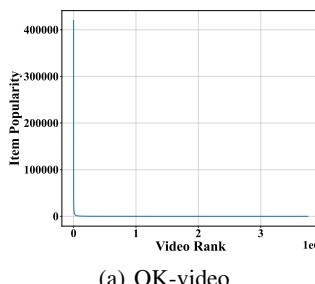

(a) QK-video

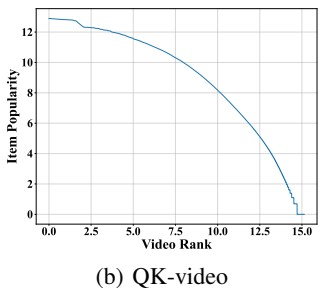

(b) QK-video

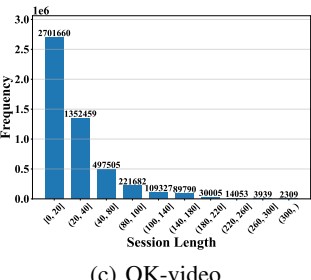

(c) QK-video

Figure 2: Item distribution. (a) and (b) are item popularity plotted in raw and log-log scales; (c) is the item session length distribution.

been desensitized for privacy issues. *User age* has been split into bins, with each bin representing a 10-year period.

The format of each instance in QK/QB-article is {*user ID, item ID, click, like, share, follow, read, favorite, click_count, like_count, comment_count, exposure_count, read_percentage, category_second, category_first, item_score1, item_score2, item_score3, read_time*}. The suffix "*∗_count*" denotes the total number of ∗ actions per article. *read_percentage* denotes how much percentage the user has read the article. *category_first* and *category_second* are categories of the article, where "*_first*" is the coarse-grained category (e.g. sports, entertainment, military, etc) and "*_second*" is the fine-grained category (e.g. NBA, World Cup, Kobe, etc.). *item_score1, item_score2, item_score3* denote the quality of the item by different scoring system. *read_time* is the duration of reading.

## 3   Experimental Evaluation

In this section, we describe the applications of Tenrec in ten distinct recommendation tasks. We briefly introduce each task and report results of popular or state-of-the-art baselines. We implemented key baselines by referring to the official code, code of DeepCTR[∗] and Recbole[∗].

### 3.1   CTR Prediction

CTR prediction is a classical recommendation task where its goal is to predict whether or not a user is going to click a recommended item. We perform this task on the sampling version of the QK-video dataset with 1 million randomly drawn users, referred to as QK-video-1M. More results of the full QK-video dataset are present in Appendix Table 2.

The reason to perform sampling is simply because searching hyper-parameters for all baselines of the ten tasks on the original dataset requires too much compute and training time. We will create a public leaderboard for the original dataset.

**DataSet.** We keep all positive feedback and draw a portion of true negative feedback with positive/negative sampling ratio of 1 : 2. By doing this, we obtain in total 1,948,388 items and 86,642,580 interactions with 96.7% sparsity. Then we split the data into 8:1:1 as the training set, validation set, and testing set.[∗] The training example consists of the following features: user ID, item ID, gender, age, video category, and user's past 10 clicked items. We apply embedding for categorical features.

**Baselines and hyper-parameters.** There are a bunch of deep learning baselines for CTR prediction. Among them, Wide & Deep  [3], DeepFM [24], NeuralFM [15], xDeepFM [24], Attention FM (AFM) [50], DCN [47] and DCNv2 [48] are some of the most well-known and powerful baselines in literature [62, 48, 39]. A recent CTR benchmark [62] shows that many recent models (e.g. InterHAt [23], AFN+ [5], and LorentzFM [52]) do not perform significantly better than these popular baselines.

---

[∗]https://github.com/shenweichen/DeepCTR-Torch

[∗]https://recbole.io

[∗]We use the 8:1:1 data splitting strategy for all tasks in this paper.

Table 2: Results for CTR prediction.

| Model | AUC | Logloss |
|---|---|---|
| Wide & Deep | 0.7919 | 0.5100 |
| DeepFM | 0.7930 | 0.5083 |
| NFM | 0.7957 | 0.5080 |
| xDeepFM | 0.7931 | 0.5081 |
| AFM | 0.7928 | 0.5090 |
| DCN | 0.7927 | 0.5092 |
| DCNv2 | 0.7932 | 0.5085 |

Table 3: Results for SBR.

| Model | HR@20 | NDCG@20 |
|---|---|---|
| GRU4Rec | 0.04882 | 0.0192 |
| NextItNet | 0.05112 | 0.0199 |
| SASRec | 0.05017 | 0.0194 |
| BERT4Rec | 0.04902 | 0.0185 |

We perform hyper-parameters search on the validation set by evaluating embedding size (denoted as $d$ throughout this paper) in $\{16, 32, 64, 128\}$, hidden units (denoted as $f$) in $\{64, 128, 256\}$, and learning rate (denoted as $\eta$) in $\{1e{-}3,\ 5e{-}4,\ 1e{-}4,\ 5e{-}5,\ 1e{-}5\}$. Finally, we set $\eta$ to $5e{-}5$, $d$ to 32 for all methods, set $f$ to 256 for DeepFM, NFM, Wide & Deep and xDeepFM, and set the attention factor to 8 for AFM. As for batch size (denoted as $b$), it performs in general slightly better by using a larger one. We set it to 4096 for all models. We find regularization $\lambda$ has no obvious effects on the results, probably due to much larger training examples, so we set it to $5e{-}5$. We set the layer number $h$ to 1 for AFM and 2 for other models according to the optimal results in the validation set.

**Results.** Table 2 shows the results of different methods on the QK-video-1M dataset in terms of Area Under Curve (AUC) [32]. We observe that in general these CTR models perform very similarly. By contrast, NFM performs the best whereas Wide & Deep performs the worst with around 0.5% disparity.

### 3.2 Session-based Recommendation

Session-based recommendation (SBR), a.k.a. sequential recommendation, aims to predicting the next item given a sequence of previous interacted items in the session, One key feature of SBR is that the interaction orders are explicitly modeled during training, which often yields better top-$N$ results.

**DataSet.** We report baseline results evaluated on QK-video-1M here. More results of the full QK-video dataset are present in Appendix Table 3. Following the common practice [56], we simply filter out sessions with length shorter than 10. Given that the average session length is 28.34, we set the maximum session lengths to 30. Session length less than 30 will be padded with zero, otherwise only recent 30 interactions are kept. After pre-processing, we obtain 928,562 users, 1,189,341 items and 37,823,609 clicking interactions. We keep the last item in the session for testing, the second to last for validating, and the remaining for training.

**Baselines and hyper-parameters.** We verify Tenrec by using four highly cited baselines: RNN-based GRU4Rec [16, 42], CNN-based NextItNet [56], self-attention-based SASRec [19] and BERT4Rec [38]. In the original paper, these models adopted different loss functions, sampling methods and data augmentations [42], which are not comparable when evaluating network architectures. Hence, to perform a rigorous comparison, we apply the standard autoregressive [56, 54] training fashion with the cross entropy loss and softmax function for GRU4Rec, NextItNet and SASRec — i.e. only the network architectures of them are different. We train BERT4Rec with the original mask token loss, which is used to compare with SASRec since both of them apply the multi-head self-attention network architecture.

The hyper-parameters are searched similarly as above. $\eta$ is set to $5e{-}4$ for GRU4Rec and $1e{-}4$ for all other three models. $b$ and $\lambda$ are set to 32 and 0 for all models. $d$, $f$ and $h$ are set to 128, 128 and 16 for NextItNet and BERT4Rec, while they are set to 64, 64 and 8 for GRU4Rec and SASRec. This is simply because SASRec and GRU4Rec produce inferior results with larger $d$, $f$ and $h$ in the validation set. The attention head is set to 4 for SASRec and BERT4Rec, which performs a bit better than 1 and 2. We randomly mask 30% items in each session for BERT4Rec after searching in $\{10\%, 20\%, 30\%, 40\%, 50\%\}$.

**Results.** We evaluate all baseline using the standard top-$N$ ranking metrics, i.e. hit ratio (HR) [56] and normalized discounted cumulative gain (NDCG) [19]. $N$ is set to 20. Table 3 shows the results of the four baselines. The obervations are as follows: (1) The unidirectional models GRU4Rec, NextItNet

Table 4: Results for MTL.

| Model | click-AUC | like-AUC |
|---|---|---|
| Only-click | 0.7957 | / |
| Only-like | / | 0.9160 |
| ESMM | 0.7940 | 0.9110 |
| MMOE | 0.7900 | 0.9020 |
| PLE | 0.7822 | 0.9103 |

Table 5: Results of TF with and without pre-training (PT).

| Model | HR@20 | NDCG@20 |
|---|---|---|
| NextItNet w/ PT | 0.12291 | 0.0489 |
| NextItNet w/o PT | 0.11922 | 0.0473 |
| SASRec w/ PT | 0.12612 | 0.0479 |
| SASRec w/o PT | 0.11715 | 0.0445 |

and SASRec offer better results than the bidirectional BERT4Rec on HR@20 and NDCG@20. This is consistent with many recent works [60, 26, 10]. (2) With the same training manner, the three unidirectional models perform similarly — NextItNet with temporal CNN architecture perform slightly better than SASRec and GRU4Rec. Our results here are different from many previous publications [38, 27, 51] where the best performed network architecture easily obtains over 50% improvements over classical baselines.

## 3.3 Multi-task Learning for Recommendation

Multi-task learning (MTL) aims to learn two or more tasks simultaneously while maximizing performance on one or all of them. Here, we attempt to model user preference of both clicks and likes rather than only one of them. We use the same dataset and splitting strategy as described for CTR prediction. The difference is that we have two output objectives for MTL with one for click and the other for like. Given that Tenrec contains many types of user feedback, one can exploit more objectives to construct more challenging MTL tasks, e.g. three-, four- or even six-task (i.e. by combing clicks, likes, shares, follows, reads, favorites together) learning using QK-article.

**Baselines and hyper-parameters.** We evaluate two powerful MTL baselines on Tenrec, namely, MMOE [28] and ESMM [29]. In addition, we also present the results of single task learning by only optimizing like or click objective. We set $\eta$, $d$, $f$, $b$ and $h$ to $1e-4$, 32, 128, 4096, 2 respectively after hyper-parameter searching.

**Results.** Table 4 shows the results of four methods on the QK-video-1M dataset in terms of Area Under Curve(AUC) [32]. As we see, ESMM performs better than MMOE for both click and like predictions. MMOE does not notably outperform the single objective optimization (SOO). Despite that, MMOE can achieve a good trade-off for two or more objectives simultaneously whereas SOO focuses only on one objective.

## 3.4 Transfer Learning for Recommendation

Transfer learning (TF) — by first pre-training and then fine-tuning — has become the de facto practice in NLP [12] and CV [17]. However, it remains unknown what is the best way to perform TF for the recommendation task [37]. In this section, we simply explore a basic way by first pre-training a SBR model (i.e. NextItNet and SASRec) in the source domain, and then transferring parameters of its hidden layer (i.e. CNN and self-attention) to the same model (with other parameters initialized randomly) in the target domain. We will study other types of transfer learning by considering data overlapping in the following sections.

**Dataset, baselines, and hyper-parameters.** We use the same dataset in the SBR task as the source dataset, and QB-video clicking feedback as the target dataset. Regarding baseline models, we evaluate the TF effects by using NextItNet and SASRec. Except $\eta$, other hyper-parameters are set exactly the same as described in the SBR task. $\eta$ is set to $1e-4$ and $5e-4$ for SASRec and NextItNet on QB-video.

**Results.** Table 5 shows the comparison results with & without pre-training. The key observation is that both NextItNet and SASRec produce better top-N results with pre-training. This suggests that parameters of the hidden layers learned from a large training dataset can be used as good initialization for similar recommendation tasks when they have insufficient training data.

Table 6: Results of user profile prediction.

| Model | Age-ACC | Gender-ACC |
|---|---|---|
| DNN | 0.67875 | 0.88531 |
| PeterRec w/o PT | 0.68671 | 0.88871 |
| PeterRec w/ PT | 0.69712 | 0.90036 |
| BERT4Rec w/o PT | 0.69151 | 0.89856 |
| BERT4Rec w/ PT | 0.69903 | 0.90082 |

Table 7: Results of cold-start recommendation.

| Model | HR@20 | NDCG@20 |
|---|---|---|
| PeterRec w/o PT | 0.03571 | 0.0194 |
| PeterRec w/ PT | 0.04412 | 0.0221 |
| BERT4Rec w/o PT | 0.03555 | 0.0192 |
| BERT4Rec w/ PT | 0.04963 | 0.0239 |

## 3.5 User Profile Prediction

User profiles are important features for personalized RS, especially for recommendations of cold/new users. Recently, [55, 2, 4, 36] demonstrated that user profiles could be predicted with high accuracy by modeling their clicking behaviors that are collected from platforms where they have more behaviors.

**Dataset, baselines, and hyper-parameters.** We conduct experiments on the QK-video-1M dataset. First, we remove instances without user profile features, resulting in 739,737 instances with the gender feature and 741,652 instances with the age feature. We split each dataset into 8:1:1 as the training set, validation set and testing set. We evaluate five baseline models for this task, namely, the standard DNN model, PeterRec [55] and BERT4Rec with and without pre-training. The pre-training and fine-tuning framework of PeterRec and BERT4Rec strictly follow [55]. Note that for PeterRec, we use the unidirectional NextItNet as the backbone , whereas BERT4Rec is bidirectional. $\eta$ is set to $1e-4, 5e-5, 1e-4$ for DNN, PeterRec and BERT4Rec respectively. Other hyper-parameters are set the same as in Section 3.2.

**Results.** Table 6 shows the results of the five baseline models in terms of the standard classification accuracy (ACC). First, PeterRec and BERT4Rec outperform DNN, indicating that the CNN and self-attention networks are more powerful when modeling user behavior sequences. Second, PeterRec and BERT4Rec with pre-training work better than themselves trained from scratch.

## 3.6 Cold-start Recommendation

Cold start is an important yet unsolved challenge for the recommendation task. A main advantage of Tenrec is that both user overlap and item overlap information is available. Here, we mainly investigate the cold-user problem by applying transfer learning. Unlike Section 3.4, both the embedding and the hidden layers can be transferred for the overlapped users.

**Dataset, baselines, and hyper-parameters.** We treat the QK-video as the source dataset and QK-article as the target dataset. In practice, there are several different cold user recommendation settings. For example, users tend to have very few clicking interactions in most advertisement recommender systems, while both warm and cold users could co-exist in other regular recommender systems. Hence, we perform evaluation for several different settings. Since cold users have been removed in our data pre-processing stage, here we simulate a simple cold user scenario by extracting his/her recent 5 interactions from these overlapped users between QK-video and QK-article. The training, validation and testing set is split into 8:1:1. We have present many additional results for other cold user settings in Table 5 of the Appendix. We use PeterRec and BERT4Rec as the baseline given their state-of-the-art performance in literature [55, 57, 2, 38]. To be specific, we first perform self-supervised pre-training on all user sequence behaviors in the QK-video dataset, then fine-tune the model in interactions of these overlapped users between QK-video and the QK-article datasets to realize the transfer learning. More details are given in [55]. Except $\eta$ in the fine-tuning stage, all hyper-parameters are set exactly the same as described in the SBR task. We set $\eta$ to $1e-3$ and $5e-3$ for BERT4Rec and PeterRec respectively during fine-tuning.

**Results.** Table 7 has shown the results of cold-user recommendation. First, we find that both PeterRec and BERT4Rec yield notable improvements with pre-training. Second, BERT4Rec with pre-training shows better results than PeterRec. This is consistent with studies in the NLP field, where bidirectional encoder enables better transfer learning than the unidirectional encoder.

Table 8: Results of LL-based cross domain recommendation. NDCG@20 is the evaluation metric.

| Model | Task1 | Task2 | Task3 | Task4 |
|---|---|---|---|---|
| Conure-NextItNet w/o PT | - | 0.0087 | 0.0162 | 0.0931 |
| Conure-SASRec w/o PT | - | 0.0081 | 0.0160 | 0.0902 |
| Conure-NextItNet | 0.0177 | 0.0095 | 0.0167 | 0.1074 |
| Conure-SASRec | 0.0172 | 0.0086 | 0.0166 | 0.0959 |

Table 9: Results of model compression. 'Cp' is the CpRec framework. Para. is the number of parameters in millions (M).

| Model | Para. | HR@20 | NDCG@20 |
|---|---|---|---|
| NextItNet | 305M | 0.05112 | 0.0199 |
| Cp-NextItNet | 204M | 0.05001 | 0.0195 |
| SASRec | 153M | 0.05017 | 0.0194 |
| Cp-SASRec | 107M | 0.04902 | 0.0191 |

## 3.7 Lifelong User Representation Learning

When transferring a neural recommendation model from one domain to another, parameters trained for the beginning task tend to be modified to adapt to the new task. As a result, the recommendation model will lose the ability to serve the original task again, termed as catastrophic forgetting [20]. [57] proposed the first 'one model to serve all' learning paradigm which aims to build a universal user representation (UR) model by using only one backbone network. In this section, we study lifelong learning (LL) by transferring user preference across the four scenarios, i.e. from QK-video to QK-article to QB-video to QB-article.

**Dataset, baselines, and hyper-parameters.** For GPU memory issues, we randomly draw 50% users from QK-video-1M as the dataset for task 1. Then we use QK-article, QB-video and QB-article for the following tasks. Given that TF and LL are more favorable to the data scarcity scenario, we process QK-article to remain at most three interactions for each user. Regarding QB-video and QB-article, we keep their original datasets because the amount of users and clicks is much smaller. Conure [57] is used as the baseline model with NextItNet and SASRec as the backbone networks. For the comparison purpose, we report results of Conure for task 2, 3, 4 without pre-training (PT) of past tasks. Model-agnostic hyper-parameters are searched similarly as before. The pruning rates are set to 60%, 33%, 25% for task 1, 2 and 3 respectively.

**Results.** Table 8 shows the recommendation results with continually learned user representations. It can be clearly seen that Conure has offered performance improvements on task 2, 3 and 4 due to PT from the past tasks. For example, Conure-NextItNet has increased NDCG@20 from 0.0081 to 0.0095 on task 2, from 0.0160 to 0.0167 on task 3, and from 0.0902 to 0.1074 on task 4.

## 3.8 Model Compression

Model Compression enables the deployment of large neural models into limited-capacity devices, such as GPU and TPU (tensor processing unit). For RS models, the number of parameters in the embedding layer could easily reach hundreds of millions to billions level. For example, [25] recently designed a super large recommendation model with up to 100 trillion parameters.

**Dataset, baselines, and hyper-parameters.** We perform parameter compression for the SBR models and use the same dataset as in Section 3.2. Despite significant research and practical value, very few work have investigated parameter compression techniques for recommendation task. Here we report results of the CpRec [40] framework, a state-of-the-art baseline in literature. We instantiate CpRec with NextItNet and SASRec as the backbone models. Hyper-parameters are set exactly the same as in Section 3.2. We partition the item set into 3 clusters with partition ratios as $25\% : 50\% : 25\%$ ranked by popularity according to [40].

**Results.** Table 9 shows that CpRec compresses NextItNet and SASRec to two thirds of their original sizes with around 2% accuracy drop.

Table 10: Results of model training speedup. 'Stack' denotes the StackRec framework.

| Model | Time | HR@20 | NDCG@20 |
|---|---|---|---|
| NextItNet | 66880s | 0.05112 | 0.0199 |
| Stack-NextItNet | 24320s | 0.05090 | 0.0202 |
| Stack64-NextItNet | 29380s | 0.05215 | 0.0200 |
| SASRec | 45040s | 0.05017 | 0.0194 |
| Stack-SASRec | 31528s | 0.05080 | 0.0196 |

Table 11: Results of model inference speedup. 'Skip' means the SkipRec framework.

| Model | Time | HR@20 | NDCG@20 |
|---|---|---|---|
| NextItNet | 308s | 0.11922 | 0.0473 |
| Skip-NextItNet | 236s | 0.12158 | 0.0472 |
| Skip32-NextItNet | 300s | 0.12439 | 0.0484 |
| SASRec | 305s | 0.11715 | 0.0445 |
| Skip-SASRec | 206s | 0.11086 | 0.0431 |

### 3.9 Model Training Speedup

This task aims to accelerate the training process of very deep recommendation models. Unlike shallow CTR models, SBR models can be much deeper. Recently, [45] revealed that SBR models like NextItNet and SASRec could be deepened up to 100 layers for their best results.[*] To accelerate the training process, they proposed StackRec, which learns a shallow model first and then copy these shallow layers as top layers of a deep model. Similarly, we evaluate StackRec by using NextItNet and SASRec as the backbone. Dataset and all-hyper-parameters are kept consistent with Section 3.2.

**Results.** Table 10 shows the results of training acceleration. Several observations can be made. (1) StackRec remarkably reduces the training time for both NextItNet and SASRec; (2) Such training speedup does not lead to a drop in recommendation accuracy. In fact, we even find that Stack64-NextItNet with a 64-layer NextItNet is trained $2\times$ faster than the standard NextItNet with 16 layers.

### 3.10 Model Inference Speedup

As the network goes deeper, a real problem arises: the inference cost increases largely as well, resulting in high latency for online services. [2] showed that users in the recommendation model can be categoried into hard users and easy users, where recommending items to easy users does not pass through the whole network. As a result, authors proposed SkipRec, which adaptively decides which layer is required for which user during model inference phase. We verify the effect of model inference acceleration in QB-video. We evaluate SkipRec by assigning NextItNet and SASRec as the backbones. Data pre-processing and hyper-parameters are set the same as Section 3.2.

**Results.** Table 11 shows the effect of SkipRec on QB-video. We see that the skipping policy in SkipRec can largely speedup the inference time of the SBR models e.g. around 23% for NextItNet and 32% for SASRec. In particular, SkipRec32-NextItNet with a 32-layer NextItNet is still faster than the original NextItNet with 16 layers. Moreover, the recommendation accuracy of SkipRec keeps on par with its original network.

## 4 Conclusions

We present Tenrec, one of the largest and most versatile recommendation datasets, covering multiple real-world scenarios with various types of user feedback. To show its broad utility, we study it on ten different recommendation tasks and benchmark state-of-the-art neural models in literature. We expose codes, datasets and per-task leaderboards to facilitate research in the recommendation community, and hope Tenrec becomes a standardized benchmark to evaluate the progress of these recommendation

---

[*]The authors also released a very large-scale SBR dataset, called Video-6M, which can be used to evaluate very deep (128-layer) RS modes.

tasks. Due to space limitation, we have not explored its full application potential. In the future, we plan to (1) investigate Tenrec for more real-world recommendation scenarios, such as cross-domain recommendation with overlapped items [35], feedback-based transfer learning (e.g. predicting likes and shares based on clicks) [57], and item recommendation with negative sampling [53, 59]; (2) release future versions of Tenrec with data that contains item modality information, such as the article title, description, and the raw video contents so as to facilitate multi-modal recommendation [46, 58][*].

## Acknowledgement

This work is supported by the Research Center for Industries of the Future (No. WU2022C030) and Shenzhen Basic Research Foundation (No. JCYJ20210324115614039 and No. JCYJ20200109113441941).

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
