# OpenReview forum: "Tenrec: A Large-scale Multipurpose Benchmark Dataset for Recommender Systems"
_NeurIPS.cc/2022/Track/Datasets_and_Benchmarks — NeurIPS 2022 Datasets and Benchmarks _

### Official Review · Reviewer_nC2L · 2022-07-13
**A very valuable large-scale multipurpose benchmark dataset for recommender systems**

**Rating:** 8
**Confidence:** 3
**Correctness:** I do not see obvious errors in their …
**Clarity:** The paper is very well written.

**Strengths:**

1. The paper was very well written.

2. Proposing a real large-scale recommendation dataset containing various user/item/behavior information.
At present, there is a great lack of high-quality real large-scale datasets in the field of recommendation systems. This work will help academic research better promote actual industrial progress, which would be a great boost to the community.

3. For the proposed dataset, the authors established comprehensive benchmarks over ten popular recommendation tasks, which facilitates the subsequent research based on this dataset. The results are also well analyzed.

**Weaknesses:**

1. It's better to specify more training details of baselines, for example, the stop rules (e.g., training fixed epochs or the details of your early stopping strategy).
2. Collaborative filtering, as one of the most classical recommendation tasks, should be tested. Besides, as the proposed dataset contains some content features, it would be grateful to additionally fuse the content information beyond user feedback.
3. It seems that only ''read_percentage'' is available. Could you also release the ''watch_percentage''?

**Additional Feedback:**

Please see above.

**Documentation:**

Yes.

**Ethics:**

No.

**Relation To Prior Work:**

Clearly discussed.

**Summary And Contributions:**

This paper releases a real large-scale dataset for recommender systems. Different from previous classic datasets like Movielens, which only have limited feedback forms, the proposed dataset, i.e., Tenrec, contains multiple types of user behaviors and is collected from four scenarios of QQ (a popular social platform). The authors also comprehensively establish benchmarks of their Tenrec dataset on up to ten concrete and important recommendation tasks. Considering the severe data scarcity problems for academic research of recommender systems, I think the dataset proposed in this work is precious.

---

> ### Author Response · Authors · 2022-08-16
> **For  Reviewer nC2L**
>
> Thank you for your nice words for our paper and the helpful suggestions.
>
> **Q1: Collaborative filtering, as one of the most classical recommendation tasks, should be tested.**
>
> Thanks your great advice. Indeed, all the four datasets introduced in this paper can be used for the standard CF. We actually planned to present this task in the beginning but we found there were already too many public datasets for such tasks. Another reason is that we have present the CTR task which could also be used to evaluate various CF models under the same sampling strategy; Moreover, we report the session-based recommendation task, where we use the time information and it usually performs better than the standard CF without the time information.
>
> As you suggested, we will easily create several online leaderboards to evaluate standard CF, for example, to evaluate model architectures, to evaluate negative sampling methods, to evaluate various loss functions.
>
> **Q2: Besides, as the proposed dataset contains some content features, it would be grateful to additionally fuse the content information beyond user feedback.**
>
> We indeed planned to release some content features such as raw video, images and the article texts, but  this might cause the privacy and copy right issues. These features need very strict censoring and we have to get permission from many team leaders. It could take longer time to get the approval from the leaders and the legal departments in the company.
>
> **Q3：It seems that only ''read_percentage'' is available. Could you also release the ''watch_percentage''?**
>
> Yes, we can only access these characteristics in the database. There are some other features we hoped to release but everything made publicly available needs very strict censoring, discussion and approval from many bosses.
>
> **Q4: It's better to specify more training details of baselines, for example, the stop rules (e.g., training fixed epochs or the details of your early stopping strategy).**
>
> Thanks for your nice suggestion. The stop rules are set following some common practice according to the performance in the validation test. More specifically, we save the checkpoints for all iteration and used the best one shown in the validation set to evaluate the testing set.

---

### Official Review · Reviewer_DWU2 · 2022-07-13
**This paper provides a publicly available data collection for recommender system that makes up for the shortcomings of the existing classic datasets and is therefore of great value.**

**Rating:** 7
**Confidence:** 4
**Correctness:** Yes

**Strengths:**

(1)	The proposed datasets make up for the shortcomings of the existing public datasets;

(2) This paper conducts experiments on ten distinct recommendation tasks for illustrating the usages of the proposed datasets.

**Weaknesses:**

(1)	Causality-inspired recommendation has gained much attention in recent years, this paper lacks discussion and illustration for this line of literature.

(2)	The examples/tasks given in Section 3.8-3.10 are not representative tasks for recommender systems, they are more of some model storage and speedup tricks.


**Additional Feedback:**

Error bars in the experimental results should be reported.

**Clarity:**

This paper is well written. Here are some minor comments.

(1)  A user’s behaviors will affect the behaviors of other users in his social network. Do the proposed datasets contain information about social networks?

(2)  Are there any missing variables (e.g., features and feedback) in the proposed datasets?

(3)  Can the proposed datasets be used as benchmarks for causality-inspired debiasing methods?

(4)  Why the metrics in Tables 2-8 are not unified?


**Documentation:**

Yes

**Relation To Prior Work:**

Yes

**Summary And Contributions:**

The benchmark datasets presented in this article are plentiful and collected from two big different real-world recommendation platforms of Tencent. These datasets have many desirable characteristics: (1) it is large-scale; (2) it includes both positive and negative user feedback; (3) it contains overlapped users and items across four different scenarios; (4) it contains various forms of feedback; (5) it includes additional features of users and items beyond the user IDs and item IDs. These characteristics enable the proposed datasets to suit different recommendation tasks. The proposed datasets make up for the shortcomings of classical datasets and have great potential.

---

> ### Author Response · Authors · 2022-08-16
> **For  Reviewer DWU2**
>
> Thanks for your positive support and helpful comments.
>
>
> **Q1: Causality-inspired recommendation has gained much attention in recent years, this paper lacks discussion and illustration for this line of literature.**
>
> Thanks, this is a great question. We have checked many causal recommendation literature, some of them studied the popularity bias. I think many datasets could be used as long as a typical long tail distribution exists in the dataset.
>
> Another direction is causal representation learning or causal environment learning [1][2][3]，these types of work focus on the generalization ability of recommendation models under two or more environments. Our datasets are well-suited since we provide four scenarios with overlapped users and items as different environments. We found many related research they often mimic a different environment, for example, treating user actions on weekdays as an environment and those on weekends as another environment.
> But honestly speaking, our authors in this paper have no practical experience for the causal recommendation task. We believe deeply mining this dataset could lead to more research tasks in recommendation. And we could build related leaderboards for various tasks which may need efforts of the whole RS community.
>
> [1] CausPref: Causal Preference Learning for Out-of-Distribution Recommendation WWW2022
>
> [2 ]Causal Representation Learning for Out-of-Distribution Recommendation WWW2022
>
> [3] Invariant Preference Learning for General Debiasing in Recommendation KDD2022
>
>
> **Q2: The examples/tasks given in Section 3.8-3.10 are not representative tasks for recommender system.**
>
> We agree with you. The main reason is that we found such types of research is becoming popular recently. In NLP and CV, large and deep models are very useful. Unfortunately, we found most recommendation models could be stacked with very few layers, e.g. only 1-3 hidden layers. More hidden layers could cause overfitting or degraded performance. But some recent research revealed that a good deep neural network could be trained with over 50 layers. This is very interesting and potentially inspire some new work. So we also hope evaluate how many deep layers can be stacked by using Tenrec. We found that we could built it with many hidden layers with 32 and 64 layers without performance drop. Maybe, it will inspire some research such as model compression or model training speedup.
>
> We believe this dataset could be used for more tasks beyond these present in this paper. We have mentioned some in the conclusion section. We believe releasing it would be very useful and we could provide related leaderboard for corresponding tasks.
>
>
> **Q3:  A user’s behaviors will affect the behaviors of other users in his social network. Do the proposed datasets contain information about social networks?**
>
> This is also a good question. We indeed could find the user’s social network but it is not allowed to publish by the company. Every feature we release need the approval of many bosses in the company.
>
> **Q4: Are there any missing variables (e.g., features and feedback) in the proposed datasets?**
>
> For real industry applications, there could be some other features and feedback. In fact, there are some features calculated based on the statistical posterior of the business, which are not allowed to use by the company (kind of business secret).
>
> **Q6: Why the metrics in Tables 2-8 are not unified?**
>
> we cannot understand this question, would you like to give more details?

---

### Official Review · Reviewer_QwW5 · 2022-07-18

**Rating:** 6
**Confidence:** 3
**Clarity:** Overall, I feel it is easy to follow …

**Strengths:**

1. The dataset will be useful towards better evaluating recommender systems for the development of the whole community. Since the tasks it involves is versatile, it has the potential to become a useful benchmark dataset for a majority of popular recommendation tasks.

2. The dataset is quite large and diverse in comparison with existing recommender systems datasets.

3. The authors have done a good job implementing several classes of baselines and the results look interesting.

**Weaknesses:**

1. The sample space of the negative sampling part seems not complete. The paper only considers the "with exposure but no user action" as the negative feedback. However, since all the exposures have been selected by the recall & rank modules, they are somewhat relevant with the users. Actually there are more negative items which are not exposed. For the CTR models, these "not-exposed" items should also been considered and showed in the training set so that the model could learn the distribution from them. With only the "with exposure but no user action" items, the sample space is not complete.

2. The comparison of the proposed benchmark with the existing benchmarks in recommender systems could be discussed more clear. The paper does not mention a lot about the competitive benchmarks. Maybe the authors could add more comparisons with the existing benchmarks in the areas of datasets size, datasets scenarios, types of tasks, etc.

3. The experimental results in CTR prediction show that the DeepFM performs even worse than W&D. This is little weird since DeepFM is the upgrade of W&D with an FM at the wide side. Maybe the authors could give more explanations for the phenomenons like this.

4. In the MTL part, the sota maybe PLE[1] rather than MMoE, the author could also add the results of PLE.

[1] Progressive layered extraction (ple): A novel multi-task learning (mtl) model for personalized recommendations. Tang, et al. 2020.

5. Some typos, such as in line 165 The → This.

**Additional Feedback:**

none.

**Correctness:**

Yes. The dataset is constructed in a sound way and the evaluation methods are appropriate.

**Documentation:**

Yes.

**Ethics:**

No.

**Relation To Prior Work:**

The comparison of the proposed benchmark with the existing benchmarks in recommender systems could be discussed more clear. The paper does not mention a lot about the competitive benchmarks. Maybe the authors could add more comparisons with the existing benchmarks in the areas of datasets size, datasets scenarios, types of tasks, etc.

**Summary And Contributions:**

The paper proposes a novel, large and versatile datasets -- Tenrec. Tenrec contains around 5 million users and 140 million interactions, which make it one of the largest datasets for recommender systems. It covers 4 different real-world scenarios with overlapped users and items, as well as various types of user feedback, including clicks, likes, shares, and follows, etc. Besides, it has not only positive user feedback, but also true negative feedback. It also contains additional features beyond the user IDs and item IDs.

The authors study the proposed datasets on 10 different recommendation tasks, including CTR Pred, session-based rec, MTL, transfer learning, user profile pred, cold-start, user representation learning, model compression, and model training/inference speedup. And they also benchmark sota models such as W&D, DeepFM, xDeepFM for CTR, and ESMM, MMoE for MTL, etc.

---

> ### Author Response · Authors · 2022-08-16
> **For  Reviewer QwW5,  thanks for your time and constructive comments. Hope our reply could solve your concern.**
>
> Thanks for your efforts and so much good advice for improving our paper. We also noticed that there might be some misunderstandings. Please take a look.
>
>
> **Q1:  However, since all the exposures have been selected by the recall & rank modules, they are somewhat relevant with the users. Actually there are more negative items which are not exposed. For the CTR models, these "not-exposed" items should also been considered and showed in the training set so that the model could learn the distribution from them. With only the "with exposure but no user action" items, the sample space is not complete.**
>
> We agree with you, this is a very worthy research question. But we guess there might be some misunderstanding. In industry, real recommender systems are always composed of many stages, roughly saying, recall and rank. For the recall stage, it is exactly as you said, sampling only true negative is not enough. The recall stage is very like many top-N recommendation tasks where the negative sampling technique matters a lot. However, CTR prediction often happens as the second stage i.e. the ranking stage. In this stage, it is common practice to use true negative (i.e., exposure but without clicking). Both for training and online serving, the CTR rank models will receive several hundreds of recalled results as input. The recalled results have the potential issue as you mentioned. But it is fine given that it is consistent for both the training and serving stages.
>
> In fact, there are very few literature discussing this sampling issue for the ranking stage.  There are indeed some papers, e.g. [1] by Facebook discussing negative sampling techniques. But these papers only focus on the recall stage rather than the ranking stage.
>
> Another perspective is that what types of negative examples should be trained is just a different evaluation setting since the aim of the CTR task in most literature is to compare RS model architectures under the same sampling strategy. I agree with you using some additional randomly chosen examples as negative may help the results further. In fact, there are several other factors that could also affect the CTR results, such as the ratio of positive and negative examples, the quality or distribution of negative samples, the mixture of true negative and randomly chosen negative. However, it is fair to compare different CTR models using the same training and evaluation data.
>
> There is another reason: in real industrial applications (e.g. at Tencent),  these "not-exposed" items will **lack some important posterior features** which are usually calculated when they are treated as useful training examples. For example, features such as the CTR score, the view rate, play rate, like rate in recent several hours are often not available for all items. We knew that for all recommendation business at Tencent, they use true negative examples to train the CTR models.  But we would like to add such results if you think it is necessary.
>
> **Please also kindly note that the criteo dataset used in DeepFM and Wide & Deep papers only use displayed  (i.e. exposures)  but non-clicked as negatives, similar as ours. (please see its description [2])** But what you suggested is a very interesting research topic and we could easily build more leaderboards to evaluate this by for example providing additional “not-exposed” items as negative. I guess it can be regarded as an open question that may require extensive evaluation for the whole RS community.
>
> [1] Embedding-based Retrieval in Facebook Search, Jui-Ting Huang et KDD 2020
>
> [2] Each row corresponds to **a display (i.e. exposures) ad served by Criteo**. Positive (clicked) and negatives (non-clicked) examples have both been subsampled at different rates in order to reduce the dataset size.
>
> **Q2:  Maybe the authors could add more comparisons with the existing benchmarks in the areas of datasets size, datasets scenarios, types of tasks, etc.**
>
> Thanks, we have made some comparisons **in the appendix in our submission version, please see Appendix Table 1 (Section B)**.
>
> **Q3: DeepFM performs even worse than W&D. This is little weird since DeepFM is the upgrade of W&D with an FM at the wide side.**
>
> Thanks so much for pointing out this.  We have run DeepFM and all other baseline models several time and revised the results in Table 2.
> In indeed, we found that in our original implementation of these CTR tasks, we evaluated the validation set every two iterations, which made some algorithms somewhat overfit. We have re-assessed them by evaluating each model per iteration and use the best model for the testing set. Please take a look at these new results.
>
> For your evaluation, **we have provided the source code, datasets of both original version and these subtasks (One can run them directly with our code and datasets: https://drive.google.com/file/d/1ss7QYHvQtfzOF1E31VrWR-_XkNHz-Jfd/view?usp=sharing )**. All of them will be attached in both the paper and leaderboard.

---

> ### Author Response · Authors · 2022-08-16
> **For Reviewer QwW5**
>
> **Q4: In the MTL part, the sota maybe PLE[1] rather than MMoE, the author could also add the results of PLE.**
>
> Thanks, **we have added the results in Table 4 (Kindly note that we did not find the official source code of PLE, we use the code in DeepCTR**.). We agree MMoE is not the best algorithm, but it is probably the most widely used baseline for multit-task learning in literature.
> We would like to add more baselines, but there are too many choices and very costly for such a large dataset with so many features and feedback
>
> For example, if we carefully search all related hyper-parameters. It may need at least 3000 experiments and aroud 1-3 years if we evaluate 10 baselines for all 10 tasks by hyper-parameter searching (e.g., learning rate {10e-5 ~ 10e-3}, regularization {10e-5 ~ 10e-3}, embedding size/hidden layer size {32, 64, 128, 256}, batch size {128, 256, 512, 1024, 2048}, dropout, layer number (1,2,4,8,16,32), attention head number （1,2,4）, sampling ratio, session length et al)
> We believe a leaderboard could help the community to identify which one is the best.

---

### Official Review · Reviewer_V8gM · 2022-07-20
**Reviews of Tenrec**

**Rating:** 6
**Confidence:** 5

**Strengths:**

1. The provided dataset Tenrec is one of the largest datasets for RS, covering a majority of recommendation scenarios and tasks(e.g. CTR prediction, top-n, sequential recommendation and so on).
2. They provide the corresponding code and will give the leaderboard of the related tasks.
3. This paper is well organized and easy to read.

**Weaknesses:**

1. The introduction of the data collection process and the application platform (QQ Kandian and QQ browser) is insufficient, which may affect the assessment of the data characteristics.
2. Lack of citation and comparison of relevant work providing datasets. Especially in the last two years some competitions/companies also provide multi-target, cross-domain or even multi-modal datasets for recommendation purposes.
3. The experimental results need to be improved. Although the code is provided and experiments are performed on 10 tasks, the majority of the experimental results are only on the sampled QK-video-10M. And some of the experimental settings and results may be questioned.
4. The extensibility of the dataset is limited. Although the dataset is large and provides auxiliary information on users and items, it lacks some rich semantic information. Specifically, for example, QK-video lacks multimodal features, and QB-article lacks content text.

**Additional Feedback:**

- Line 19-21. “Many improvements have been achieved in prior literature, however, much
20 of them performs evaluation on non-benchmark datasets…”.  Although other recommendation datasets may not be as large as Tenrec, they can be considered as benchmark datasets.
- Line 47-48. The provided dataset contains mainly ID features and does not seem to be well suited for content-based recommendations.
- Fig. 1.  Does the location of the subfigure have any special meaning, especially the QK-video subfigure.
- Fig. 2(a) and (b). Please explain these two figures more clearly.
- L116-118.  I'm confused about the reason for sampling. Without sampling, I ran CTR experiments and multi-task experiments on the full QK-video dataset using an 11G GPU.
- Line 119-120.  The original data set has a positive to negative sample ratio of 1:2.4, and I believe that additional sampling is not necessary.
- Table 2. According to my experimental results on the unsampled QK-video, AFM hardly outperforms xDeepFM by more than 1%.  I strongly encourage the authors to tune the experimental results more carefully and to add the experimental results of DCN and DCNv2.
- Line 158. Sampled softmax seems to be a more general approach. If sampled softmax is used, please provide the number of negative samples.
- I suggest that the authors use both gAUC and AUC as evaluation metrics, especially for CTR prediction and Multi-task Learning.
- Table 4. The experimental results of PLE should be added. In addition, in the experiments I did on the unsampled QK-video, the AUC of click is significantly higher than that reported in the paper. Therefore, I doubt that the authors' sampling is biased.
- Line 183-185. All samples in QK-article are click-positive samples, so QK-article is not applicable to six-task learning.


Overall, I think Tenrec is valuable to the development of RS. I encourage and expect the authors to refine the experiments and provide more solid leaderboards and the corresponding reproduction details.

**Clarity:**

This paper is very well written and I can understand it clearly and easily.
However, some details about figures(e.g. Figure 1) need to be added more clearly. Also, the github address of the code could be placed more prominently.

**Correctness:**

I think the core contribution of this paper is the multi-purpose dataset provided.
1. Empirically, the dataset is correctly constructed and corresponds to the description in the paper.
2. The raw data is provided, but the processed dataset is temporarily unavailable. Therefore, although I have some doubts about the experimental setup and results, I cannot absolutely judge the correctness.

**Documentation:**

The original dataset has been provided, as well as the corresponding presentation in the paper. However, the processed dataset used in the paper is not yet available.

In addition, I suggest that the authors provide an additional dataset introduction document (in addition to the paper).

Of course, I believe that the above issues will be resolved if this paper is published.


**Ethics:**

The dataset has been desensitized, so I am not currently aware of any moral or ethical implications of this paper.


**Relation To Prior Work:**

This paper discusses the limitations of datasets for recommendation systems and uses ImageNet and GLUE as examples to illustrate the necessity for large-scale recommendation datasets. Of course, they also mention some datasets, but there is a lack of intuitive comparison and references to some other large-scale datasets such as Criteo, Taobao, etc.


**Summary And Contributions:**

1. This paper provides a large-scale dataset Tenrec valuable for RS, containing multiple scenarios, various user feed back, user profiles, etc.  It is useful for the development of recommendation communities.
2. This paper illustrates 10 tasks to which the Tenrec dataset can be applied, and provides some of the experimental results.
3. This paper points out the lack of sufficiently large new datasets for recommender systems, which hinders the growth of the community. This motivation is very clear and worthy of recognition.

---

> ### Author Response · Authors · 2022-08-16
> **For Reviewer V8gM --- please take a look, we have revised the paper following your suggestions**
>
> Thanks for your feedback to improve our paper. Please see our replies to some key questions.
>
>
> **Q1 :  Also, the github address of the code could be placed more prominently.**
>
> Thanks for your advice, the code and dataset links will be in the abstract for the camera-ready version.
>
> **Q2: Lack of citation and comparison of relevant work providing datasets. Especially in the last two years some competitions/companies also provide multi-target, cross-domain or even multi-modal datasets for recommendation purposes.There is a lack of intuitive comparison and references to some other large-scale datasets such as Criteo, Taobao, etc.**
>
> Thanks, please kindly note that in our submission version, we have provided a related work comparison **in the appendix Table1** since there is very limited space in the main body of this paper.
>
> **Q3: The original dataset has been provided, as well as the corresponding presentation in the paper. However, the processed dataset used in the paper is not yet available.**
>
> Thanks, we have appended the processed dataset in the appendix with red color.
>
> **Q4: The introduction of the data collection process and the application platform (QQ Kandian and QQ browser) is insufficient, which may affect the assessment of the data characteristics.**
>
> Thanks,  QQ Kandian and QQ browser are two very large recommendation platforms/apps with over 100 million DAU (Daily Active User). Both of them includes items such as news articles, short videos, advertisements. As for the data collection part, we have detail them in Section 2. Please let us know if you want to know some other details. In fact, when we submited this paper, the company side hoped us to provide as few information as possible about the business since too much information (e.g., real feature names, the ratio of true negative examples) will potentially cause the disclosure of business secrets. Even the names of recommender system they hope we could not mention it. But it is fine now after many discussions with several leaders. Honestly speaking, it is very very difficult to publish data from the commerical company.
>
> **Q5: The majority of the experimental results are only on the sampled QK-video-10M. And some of the experimental settings and results may be questioned.**
>
> Thanks, in our paper, we report some additional results using the original dataset with 140 million positive interactions in the Appendix Table 2 and Table 3.
>
> It is very difficult to run all baselines on all datasets since  searching the hyperparameter for so many datasets so many tasks and so many basliens requires several thousands of experiments.
>
>
> For example, simply assuming that for each of the 10 tasks, we run 10 baseline models, and for each model we carefully search hyper-parameters (e.g., learning rate {10e-5 ~ 10e-3}, regularization {10e-5 ~ 10e-3}, embedding size/hidden layer size {32, 64, 128, 256}, batch size {128, 256, 512, 1024, 2048}, dropout, layer number, attention head number, sampling ratio, session length et al, in total there should be at least at least 30 linear combinations for these hyper-parameters).
>
> This means there will be “at least 3000 experiments” to be run on only one dataset. Running these experiments on the original large datasets could take at least 1~3 years even using 10 powerful GPU cards (excluding the implementation time for so many baseliens).
>
>
> **Q7: The extensibility of the dataset is limited. Although the dataset is large and provides auxiliary information on users and items, it lacks some rich semantic information. Specifically, for example, QK-video lacks multimodal features, and QB-article lacks content text.**
>
> Yes, we agree with you. If there are some content data for items, it could be more userful. Howeverthese contents are very sensitive. User privacy issue could be inferred from multimodal features or real content features with clicking behaviors. We have spent a lot of efforts even for releasing the current datasets (with so many features, feedback with a large scale). We had applied for the approval from about 6 leaders at Tencent and the law department as well.
>
> **Q8：The raw data is provided, but the processed dataset is temporarily unavailable.**
>
> It is availble now, please see the appendix with a red URL
>
> **Q9: Fig. 2(a) and (b). Please explain these two figures more clearly.**
>
> It is the figure describing long-tail item popularity distribution, (a) x-axis and y-axis are the item rank (in terms of popularity) and item popularity (in terms of clicking) respectively. (b) is the log-log scale. Such figures are used by many previous papers, please see [1] Figure1 (We draw it by exactly following this paper)
>
> [ 1] Improving Pairwise Learning for Item Recommendation from Implicit Feedback. Rendle et al

---

> ### Author Response · Authors · 2022-08-16
> **For Reviewer V8gM**
>
> **Q7:  I'm confused about the reason for sampling.**
>
> Thanks for correcting our description. The reason we perform sampling is mainly due to the compute resources, running time and GPU memory issues. In particular, for session-based recommendation with standard full softmax loss function (better than the sampled version and binary CE loss), It requires very large GPU memory. Using small batch size could cause accuracy degradation and much longer training time.
>
> **Q8: The original data set has a positive to negative sample ratio of 1:2.4, and I believe that additional sampling is not necessary.**
>
> We agree with you. We find some literature use 1:1 or 1:2 or other ratios. We simply use 1:2. It is a random choice without too much thinking. We could report related results if you think model results with the original data is better.
>
> **Q9: Table 2. According to my experimental results on the unsampled QK-video, AFM hardly outperforms xDeepFM by more than 1%. I strongly encourage the authors to tune the experimental results more carefully and to add the experimental results of DCN and DCNv2.**
>
> Thanks so much for pointing out this issue. We have run these models again and report new results in Table 2. We found xDeepFM overfitted a bit in previous results since in our original implementation we report the validation results every two iterations. The model became kinds of worse after another one or  two iterations. The results have been revised in the PDF file with red color. We have also provided the processed datasets for each task in the appendix with red color (Appendix Section A). Your can directly run our code now to reproduce these results.
>
> We also provided the results of AFM vs. xDeepFM using the original large dataset. Please check whether ours results are the same with yours. The results of the two models are very close.
>
> **Q10 In addition, in the experiments I did on the unsampled QK-video, the AUC of click is significantly higher than that reported in the paper.**
>
> Yes, please kindly see **Table 2 in the appendix**, which we had provided some results using the original dataset in our submission version. The accuracy is much higher than the sampled version.
>
> **Q11： Sampled softmax seems to be a more general approach. If sampled softmax is used, please provide the number of negative samples.**
>
> For session-based recommendation，we use full softmax since it shows better results than sampled version and largely better than binary cross entropy loss and NCE loss. We could provide leaderboards for different sampling methods since the impact of samplers are more higher than the network architecture. That is probably the reason why in some  papers these baselines could have very large accuracy disparity.
>
> **Q12:  All samples in QK-article are click-positive samples, so QK-article is not applicable to six-task learning.**
>
> Sorry, each dataset in this paper is in charge by a different technical team at Tencent. We cannot access, download or publish the data without the approval of team leaders and some other bosses. But we thought the current version of Tenrec has already provided a lot of useful information for so many tasks.

---

### Official Review · Reviewer_cQTo · 2022-07-22
**Review of "Tenrec: A Large-scale Multipurpose Benchmark Dataset for Recommender Systems"**

**Rating:** 6
**Confidence:** 5

**Strengths:**

I really enjoyed this dataset that it has the potential to become a useful benchmark for recommendation community since it has several strengths.
+ Compared with majority existing datsets, Tenrec has five characteristics: (1) large-scale, containing around 5 million users and 140 million interactions; (2) contains not only positive user feedback, but also true negative feedback (vs. one-class recommendation); (3) contains overlapped users and items across four different scenarios; (4) contains various types of user positive feedback, in forms of clicks, likes, shares, and follows, etc; (5) contains additional features beyond the user IDs and item IDs.
+ They verify Tenrec on ten diverse recommendation tasks by running several classical baseline models per task.

**Weaknesses:**

As mentioned above, I am very excited about this paper, however, I think that it still have some limitations need to clarify and need more details.
+ Some experiment settings and strange results for benchmark need to be clarify. See more details in **Correctness**.
+ The document about the dataset processing for specific tasks are missed, which is also important to promote reproducibility for our community.
+ The authos have no mention of some other multiple (more than 2) feedback datasets, which is very related.
+ The methods for each task is too few and more popular and strong baselines are not included.


More concerns are described in the following sections, please check them.

**Additional Feedback:**

All the comments and suggestions are provided above. This is a dataset full of potential, and if you can address my concerns and add more details about the questions above, I would like to increase my score.

**Clarity:**

The paper is well written, however, some parts are not clear. Consider giving some examples.
1. I could not find any description for Figure 1 in the main text (i.e. not explained at one place).
2. In section 2, are videos and articles mixed and displayed on the same platform? Or displayed separately? In figure 1, I think you want to mean the latter case. More introductions to scenarios in which raw data are collected should be included.


**Correctness:**

The paper is correct for the most part. The dataset construction is sound. Details of different methods and experiments are included. However, for benchmark, I still have several concerns as follows.
1. Since you have used the DeepCTR to run baselines, and I see that you copyed the class 'VarLenSparseFeat' from DeepCTR (line 41 in model/ctr/inputs.py), however, for sequence features (i.e., user’s past 10 clicked items in your paper), you embeded them separately(line 84 in utils.py), I wonder whether it is a better idea to share the same embedding with item embedding rather than separately? Could you please give some explanation?
2. Since Tenrec have timestamp, why don't split the data into the training set, validation set, and testing set according to the time, which is a common setting in industry?
3. More metrics should be evaluated, for example, GAUC and Logloss for CTR prediction task.
4. For MTL task, why only-click and only-like better than MTL methods? And what is the training data for like prediction? Is there also selection bias here like CTR and CVR? Maybe you could add more details.
5. Some strong and popular models for the tasks you mentioned can also be included, for example, DIN[1] and DIEN[2] for CTR prediction task.

[1] Deep Interest Network for Click-Through Rate Prediction, KDD 2018.
[2] Deep Interest Evolution Network for Click-Through Rate Prediction, AAAI 2019.

**Documentation:**

The benckmark codes is included in the GitHub repo. However, I recommend you could add more details (including documents and codes) about how to construct the appropriate dataset based on your four original dataset for different tasks mentioned in your paper. The way datasets are processed to serve different tasks can also make contribution to our community, and I think that it can be an important contribution for your work as well.

**Ethics:**

No. All the user ID and item ID are anonymized.

**Relation To Prior Work:**

The authors have mentioned some existing recommender systems datasets and compared them in Table 1 in Appendix.  However, there are some other datasets that contain multiple (more than 2) user feedbacks (e.g. UserBehavior(https://tianchi.aliyun.com/dataset/dataDetail?dataId=649), Ali_Display_Ad_Click(https://tianchi.aliyun.com/dataset/dataDetail?dataId=56)), which also need to be compared since one of your Tenrec's biggest merits is that it contains multiple types of positive user feedback.

**Summary And Contributions:**

The authors proposed Tenrec, a large-scale and multipurpose real-world dataset, which have several characteristics. They also examine its properties by ten recommendation tasks by running some popular or state-of-the-art baselines.

---

> ### Author Response · Authors · 2022-08-16
> **For Reviewer cQTo --- We have added related experiments as suggested, please take a look our Appendix.**
>
> Many thanks for your constructive feedback for improving our paper and chances for increasing the score.
>
> **Q1: Correctness: for sequence features (i.e., user’s past 10 clicked items in your paper), you embeded them separately (line 84 in utils.py),  I wonder whether it is a better idea to share the same embedding with item embedding rather than separately?**
>
> Thanks for the great question, as suggested, we have reported new results by considering both shared embedding and separate embedding **in Table 4 of the Appendix (with description in Section C)**. As can be seen, in general, separate embedding performs slightly better than shared embedding. We have provided the source code for both settings. Please see our github: https://github.com/yuangh-x/2022-NIPS-Tenrec/blob/master/utils.py  Please kindly note that all processed datasets for the 10 tasks can be found in our appendix with red link color. All the experiments can be run directly: https://github.com/yuangh-x/2022-NIPS-Tenrec/blob/master/main.py
>
> In fact, whether some embeddings can be shared or not is an open question. Some researchers may use shared embeddings (a similar issue fo bottom item embedding and softmax matrix embedding) while others may use separate embeddings. In general, the performance will not be largely different and as long as we ensure that all models are run with the same setting should be fine. We believe with a public dataset and leaderboard, such questions could be well addressed in the future.
>
> **Q2:  why don't split the data set according to the time.**
>
> We guess there might be a misunderstanding.  **We indeed split datasets by time**, for session-based recommendation and other related task.
>
> As for CTR prediction, we also construct examples based on time information (e..g, using the past behaviors as user features and future behaviors as target item ID.  For each user, the earliest 10 actions are treated as features, others are target item IDs).  After constructing the examples, we randomly split the training, validation and testing data by 8:1:1, **which is a common practice in a lot of literature,
>  e.g., DeepTables [1], DeepCTR [2] (6.3K stars), DeepFM and [3] (please see 3.2.2).**
>
> [1]https://deeptables.readthedocs.io/en/latest/
> [2] https://github.com/shenweichen/DeepCTR
> [3] Open Benchmarking for Click-Through Rate Prediction
>
> We agree there are more evaluation setups, but with a public dataset, researchers can easily evaluate different models and settings and create new online leaderboards if necessary.
>
> **Q3:  More metrics should be evaluated, for example, GAUC and Logloss for CTR prediction task.**
>
> Thanks for pointing out this. **We have added Logloss in Table 2 (both main body and appendix)**. The  main reason we did not show these metrics is mainly because when preparing this paper, we noticed some typical work like Google’s “Wide & Deep” and Alibaba’s DIN [1] and several recent work e.g. [2], they did not use Logloss or GAUC.   We would like to add GAUC in the online leaderboard.
>
> [1] Deep Interest Network for Click-Through Rate Prediction
> [2] Agile and Accurate CTR Prediction Model Training for Massive-Scale Online Advertising Systems. SIGMOD2021
>
> **Q4:  The methods for each task is too few and more popular and strong baselines are not included. Some strong and popular models for the tasks you mentioned can also be included, for example, DIN[1] and DIEN[2].**
>
> Thanks, we have **added the results of DIN[1] and DIEN[2] in Table 2** and results using the original large-scale datasets in the Appendix marked with red color.  The reason that we have not included all baselines in the submission version is simply because there are too many choices if we consider 10 tasks with careful hyper-paramter searching.
>
> For example, if we run **10 baselines for each of the 10 tasks**, and carefully fine-tune the hyper-parameter, including, learning rate {10e-5 ~ 10e-3}, regularization {10e-5 ~ 10e-3}, embedding size/hidden layer size {32, 64, 128, 256}, batch size {128, 256, 512, 1024, 2048}, dropout, layer number (1,2,4,8,16,32), attention head number （1,2,4）, sampling ratio, session length et al, in total there could be at least **30 linear combinations for each model**.  In other words, there will be at least **3000 experiments for each dataset**. Roughly speaking, it will take 1-3 years with 10 powerful GPUs (excluding the reimplementation time of these baselines).
>
> To evaluate the progress of RS, we believe an online leaderboards could be more helpful, the results of which will be constantly updated.
> To update the leaderboard, we would require the authors to provide source code and hyper-parameter settings. This should guarantee the reproducibility. In the paper, we mainly show results of some representative baselines rather than all baselines since the reported baselines will be surpassed by some new SOTAs soon.

---

> ### Author Response · Authors · 2022-08-16
> **For Reviewer cQTo**
>
> **Q5: The authors have no mention of some other multiple (more than 2) feedback datasets, which is very related.**
>
> We showed several datasets in appendix Table 1. Many thanks for letting us know the two Alibaba datasets, and we have added the two datasets as suggested  **in the appendix Table 1 (please check it with red color)**.
>
> **Q6: For MTL task, why only-click and only-like better than MTL methods?**
>
> MTL aims to optimize two or more objectives, but cannot guarantee they are always the best for any one of them. There is a negative transfer issue for MTL revealed by some literature. **For reproducibility, we have released all processed datasets (with red URL color) and code in the  appendix . These models can be directly run with our code.**
>
> **Q7: The document about the dataset processing for specific tasks are missed, which is also important to promote reproducibility for our community.**
>
> Thanks, we have released all processed datasets attached with source code of the ten tasks  with red URL color in the  appendix.
>
> Tenrec data link: https://drive.google.com/file/d/1R1JhdT9CHzT3qBJODz09pVpHMzShcQ7a/view
>
> Datasets for the ten reported tasks: https://drive.google.com/file/d/1ss7QYHvQtfzOF1E31VrWR-_XkNHz-Jfd/view
>
> Baseline Code: https://github.com/yuangh-x/2022-NIPS-Tenrec
>
> Data preprocessing code: https://github.com/yuangh-x/2022-NIPS-Tenrec/tree/master/Data%20Processing
>
>
>
> **Q8: In section 2, are videos and articles mixed and displayed on the same platform? Or displayed separately?**
>
> Video and news recommendation are run by different technical teams with complete different RS algorithms，including different recalls, ranks, user and item features. But they are displayed in the same platform. Also note that QK and QB are different platforms/Apps. We describe this in Section 2.
>
> **Q9: I could not find any description for Figure 1 in the main text (i.e. not explained at one place).**
>
> Sorry, we have some co-authors working on these real production systems.  They have removed some descriptions but we did not notice this in the submission version. In the beginning, the boss  even suggested us not to mention the real names of the recommender systems. We have made a lot of efforts and discussions to persuade the leaders. But there are  some features and descriptions that  are still very sensitive. We have added some basic descriptions. Please let us know if you hope to know more details.

---

> ### Author Response · Authors · 2022-08-20
> **For reviewer cQTo, we have revised this paper following your instructions.**
>
> Dear reviewer cQTo,
>
> Would you like to take a look at our rebuttal, we have revised the paper strictly following your constructive comments. Please let us know if there are  any places you think we could further improve.
>
> Thanks again.
>
> Regard

---

> > ### Comment · Reviewer_cQTo · 2022-08-20
> > **Thanks for the author's response, it's a good job.**
> >
> > Thank you for the point by point response and add more details.  Most of my concerns are largely addressed and I would like to increase my score to 6 (marginally above acceptance threshold).
> >
> > Although I agree with the authors' response on explanation about Q6, I still have some doubts about the result. Thank you for your opensource codes, and I will run it for further check.  It is a interesting phenomenon, if all MTL methods did not outperformed the only-click and only-like, there is no point in designing MTL methods.
> >
> > For Q8, as you explained, video and news recommendation are displayed in the same platform, so how to combine and display the rank list from the two recommender systems (video and news) is also an interesting problem (i.e, maybe like a rerank problem?).
> >
> > I agree that more baselines and results need too many time and it takes the whole community to work together.
> >
> > Thank you for adding more experiments and expannations, and open sourece the data processing code. I believe it is a good job and will help recommendation community.

---

> > > ### Author Response · Authors · 2022-08-21
> > > **Thanks so much for increasing the score and so many good comments!**
> > >
> > > Dear reviewer,
> > >
> > > thanks so much for helping us improve the quality of this paper
> > >
> > > Q1: Regarding the effectiveness of MTL
> > >
> > > At Tencent, we designed two objectives for video recommendation business: one is whether there is a clicking behavior, and the other is how long this video has been played.  It has a similar issue, i.e., there is no obvious improvements if we only consider one metric but it indeed helps if we hope the recommended items meet both objectives. Both clicking and playing time matters for the real production system. For example, if we only consider the clicking behavior, it is very likely that some of the recommended items have very attactive title or cover image (tempting users to click), but their contents can be low quality. Users tend to skip these videos when they are played for only several seconds.  Hence, we often use another objective i.e., the playing time, to filter such items.
> > >
> > > We noticed that in the MMOE paper, the AUC improvements of MTL is also very minor, around 0.1 % (Table 1) and < 0.2% (Table 2) compared with the single task learning paradigm. There is one metric (AUC/Marital Sta) showing even worse results.
> > >
> > >  Q8: Regarding how to display the rank list from the two recommender systems
> > >
> > > It is very interesting if we could design some algorithms to determine which item (top-ranked news or top-ranked videos)）ought to be placed in the first position. Given that the news and video recommendation are regarded as two independent business (in both QK and QB) and run by two technical teams, there is no competetion between the two recommendation algorithms. The solution is to provide some fixed templates or slots, which determines whether this position is given to the news or a video.

---

### Official Review · Reviewer_W64h · 2022-07-27
**A Large-scale Multipurpose Benchmark Dataset for Recommender Systems**

**Rating:** 6
**Confidence:** 5
**Clarity:** This paper is well-written and easy t…

**Strengths:**

Different from existing recommendation datasets, the proposed Tenrec have following strengths:
1) Large-scale real-world dataset, which contains 5 million users and 140 million interactions;
2) Containing multiple types of behavior and true negative feedback, which could be utilized to support CTR scenario.
3) Consisting of overlapped users/items, which could support the CDR and TF research;
4) Including additional user and item features, which could be used for side information aware methods.

Owing to these merits, I think this dataset have potential to become a widely used benchmark dataset in the future, which could advance the recommendation research in the academia.


**Weaknesses:**

Here are some concerns about this paper:
1) In Table 1, why the #exposure of QK-article and QB-article are missed?
2) Section 3.8, 3.9, 3.10 is weakly related to the contributions of this dataset. The authors should choose the tasks that could better associate with the merits of the Tenrec, such as utilizing true positive negative feedback and additional user and item features. Although model depression and acceleration are also important topics, it is more appropriate to put them to the appendix or as the future work.
3) The authors did not clearly illustrate what kind of user behavior they used for some tasks like session-based recommendation, transfer learning for recommendation ...
4) Given the fact that the SASRec performs better than BERT4Rec in Table 3, why the authors still use BERT4Rec for cold-start recommendation? It is hard to convince me that "bidirectional encoder enables better transfer learning than the unidirectional encoder" without comparing with SASRec.
5) In line 234, the authors said that "We only study item recommendation for users who have interactions less than 5 in the QK-article dataset ...", however, in the dataset description part, the author just said "the requirement that each user had at least 5 video clicking behaviors ..." "We perform similar data extraction strategy for QK-article, QB-video, and QB-article.", so the question is, why does the QK-article dataset have interactions less than 5? The authors should make it more clear.
6) To summary, I think the authors should focus on the tasks that are highly related to the five characteristics described by them. If possible, please include more baselines to compare.

**Additional Feedback:**

Some typos:
1) Line 230, "Tenure" should be "Tenrec";
2) Line 231, "by apply" should be "by applying".

**Correctness:**

The claims seem correct for me. The dataset is constructed in a sound way. And most of the evaluation methods and design seems appropriate and are performed correctly.

**Documentation:**

The documentations provide enough details about the data collection and organization. For benchmark, the authors have provided sufficient details for reproducibility.

**Ethics:**

There is no ethical concerns.

**Relation To Prior Work:**

This paper has clear discussion about the difference between Tenrec and other recommendation datasets.

**Summary And Contributions:**

This paper proposed a large-scale multipurpose benchmark dataset for recommender systems and verified it on 10 recommendation tasks.

---

> ### Author Response · Authors · 2022-08-16
> **For Reviewer W64h**
>
> Thinks for your great advice and helpful feedback on our work. Some of your questions are replied as below.
>
> **Q1: In Table 1, why the #exposure of QK-article and QB-article are missed?**
>
> Thanks, this is simply because the video and news data belong to different teams at Tencent. To release any information, we need to get the permission from several team leaders. With the exposure information, one could easily infer the quality of their recommendation algorithms and they probably do not want to provide such data for business secret reasons.
>
> **Q2: Section 3.8, 3.9, 3.10 is weakly related to the contributions of this dataset. The authors should choose the tasks that could better associate with the merits of the Tenrec, such as utilizing true positive negative feedback and additional user and item features.**
>
> Thanks, the main reason is that we found with Tenrec data, we could build sequential recommendation models with very deep neural networks (e.g. up to 64 layers without accuracy drop). We found that neural recommendation models with very deep （> 10）hidden layers easily overfit or cause performance degradation on many public datasets. That is one character here we want to show. With Tenrec, we may build very deep models for recommendation tasks.
>
> With Tenrec, we can develop more tasks and create more leaderboards. These tasks shown in this paper can be regarded as an example. For example, we did not show the standard top-N recommendation (because there are already many public datasets), multi-task learning with three or more feedback, negative sampling task, cold-item recommendation, etc. We hope this dataset could inspire more new tasks and we would create corresponding online leaderboards in the future.
>
> **Q3. The authors did not clearly illustrate what kind of user behavior they used for some tasks like session-based recommendation, transfer learning for recommendation.**
>
> Sorry for missing such information. Such information can be obtained by our description. For example, we mentioned there are 37,823,609 interactions. Table 1 shows that only clicking behaviors has such a scale. We have revised here with red color in section 3.2 and 3.4 and have released all processed datasets for the ten tasks (see Appendix section A with red URL).
>
> **Q4: Given the fact that the SASRec performs better than BERT4Rec in Table 3, why the authors still use BERT4Rec for cold-start recommendation?**
>
> This is a great question. First, sequential (session-based/next item) recommendation in Table 3 is essentially a generation task, where SASRec (autoregressive generation) is often better than BERT4Rec. This has been demonstrated in a lot of literature (also in the NLP tasks), although the original paper showed that BERT4Rec was better. In fact, in the NLP domain, it is widely accepted that BERT is not good at the generation tasks. In constrast, BERT is more often used as a powerful encoder for the semantic understanding task rather than token generation tasks.
>
> For cold-start recommendation (as well as the user profile prediction tasks), it is essentially an sequence-level understanding/classification task where bidirectional encoder is better-suited than the unidirectional transformer.
>
> In summary, we use BERT rather than SASRec for these understanding tasks is simply because bidirectional encoder is more popular for these types of tasks. It could happen with good tricks unidirectional transformer could be better than bidirectional BERT, but that is fine, we could simply show these new results on an online leaderboard and this is the real purpose of this paper. Note that we indeed used SASRec and NextItNet for the lifelong learning task simply because in the original paper the author used the unidirectional models. It also works well (maybe better) using a BERT or bidirectional NextItNet as the backbones.
>
> **Q5: why does the QK-article dataset have interactions less than 5?**
>
> Thanks for correcting our statement error.  Indeed, cold users were removed in the first step. Here, we simply simulate a cold user setting by extracting the recent 5 behaviors for each user. In practice, there are many different cold-start scenarios, for example, in some recommender systems where both cold-users and warm-users co-exist, we have added such results in Table 5 of the appendex. **We have revised our paper marked with red color for both the main body part and appendix (Section C).**

---

### Author Response · Authors · 2022-08-03
**Dear reviewers, thanks for all your constructive comments and nice words. we hope to respond to one key question -- regarding adding more baselines or evaluating more datasets.**

Honestly speaking, adding all related baselines for each task is very expensive for 10 tasks, there are too many choices, please see our explanation as below (kindly note that we have added most results suggested by our reviewers):

Simply assuming that for each of the **10 tasks**, we run **10 popular baseline** models, and for each model we need carefully search hyper-parameters (e.g., learning rate {10e-5 ~ 10e-3}, regularization {10e-5 ~ 10e-3}, embedding size/hidden layer size {32, 64, 128, 256}, batch size {128, 256, 512, 1024, 2048}, dropout, layer number (1,2,4,8,16,32),  attention head number （1,2,4）, sampling ratio, session length et al, in total there should be at least at least **30 linear combinations** for each model).  Also note there are **4 datasets** in total if we run experiments on all of them.

This means there will be **at least 3000 experiments** to be run. Since our dataset is much larger than those used in much previous literature，if we run all experiments with the original dataset (with 140 million interactions, training a deep RS model used in this paper may require 24 -  72 hours, depending on the batch size, sampling ratio, iterations，algorithms etc.). Even under ideal conditions, it takes **at least 1 ~ 3 years (excluding the time to reproduce these baselines) using 10 powerful GPU cards**. We believe it is very difficult to ensure that all 100 baselines with 3000 experiments are perfectly evaluated by one paper. Such work should be done by the whole RS community and our contribution is to provide the datasets and public leaderboards – the main purpose of this paper.

We hope to clarify the misunderstanding that this is not a benchmark paper that evaluates or questions some papers or some recent advances in recommender systems. Instead, the focus is to introduce a new dataset for the community and this new dataset promises to become a valuable benchmark for various RS tasks with the efforts of the whole community. As mentioned, to evaluate the progress, we will create corresponding public leaderboards, which will be constantly updated. This paper mainly shows results of some representative baselines rather than all related baselines. These reported results in our paper should be soon outperformed once there is a public leaderboard.

For updating the leaderboard, we would require the authors to provide source code and hyper-parameter settings. This should guarantee the reproducibility. In other words, **the key purpose of this paper is to improve the reproducibility of the RS community by introducing a large-scale dataset and public leaderboards, rather than focusing on which model is the best in each task**.  All reported baselines should be soon surpassed by some new SOTAs.

Please kindly note that, **we have supplemented new experiments in the Appendix and main body with red color as suggested by some of our reviewers**.

---

### Author Response · Authors · 2022-08-20
**Dear all reviewers, we have attached all related datasets, all codes, and all required baselines and experiments as suggested, please take a look.**

We thank all reviewers for their constructive comments, which are very important for improving our paper. All code and datasets are provided in our **Supplementary Material (Appendix A)**. To be specific,

Tenrec data link:
https://drive.google.com/file/d/1R1JhdT9CHzT3qBJODz09pVpHMzShcQ7a/view

Datasets for the ten reported tasks:
https://drive.google.com/file/d/1ss7QYHvQtfzOF1E31VrWR-_XkNHz-Jfd/view

Baseline Code: https://github.com/yuangh-x/2022-NIPS-Tenrec

Data preprocessing code: https://github.com/yuangh-x/2022-NIPS-Tenrec/tree/master/Data%20Processing

We have supplemented more experiments, including new baselines, running models on the original large datasets, which are marked with red color in both the main body and appendix.

Best regards

---

### Meta-Review · Area_Chair_W1mj · 2022-09-09

**Recommendation:** Accept
**Confidence:** 4

**Metareview:**

In the last few years hyperscalers such Meta and Google have stated, more or less plainly, that the vast majority of their "AI Cycles" in the datacenter are devoted to various forms of recommendation tasks. Unfortunately, it has been hard to perform research on this problem, outside of those companies, due to the private and proprietary nature of the data. Large open benchmark sets have been limited to more narrow domains such as movie reviews. This paper offers a more representative large-scale real-world dataset which contains 5 million users and 140 million interactions.  The authors have gone to great lengths to address reviewers feedback, and, following their rebuttals, the reviews of this paper are now uniformly positive. This paper should be accepted.

---

### Decision · Program_Chairs · 2022-09-16

Accept